# Deimination and Peptidylarginine Deiminases in Skin Physiology and Diseases

**DOI:** 10.3390/ijms21020566

**Published:** 2020-01-15

**Authors:** Marie-Claire Méchin, Hidenari Takahara, Michel Simon

**Affiliations:** 1UDEAR, Institut National de la Santé Et de la Recherche Médicale, Université Paul Sabatier, Université de Toulouse Midi-Pyrénées, U1056, 31059 Toulouse, France; marie-claire.mechin@inserm.fr; 2University of Ibaraki, School of Agriculture, Ibaraki 300-0393, Japan; hidenari86ta@msn.com

**Keywords:** epidermis, peptidylarginine deiminase, citrullination, hair, alopecia, posttranslational modification, keratinocyte, differentiation

## Abstract

Deimination, also known as citrullination, corresponds to the conversion of the amino acid arginine, within a peptide sequence, into the non-standard amino acid citrulline. This post-translational modification is catalyzed by a family of calcium-dependent enzymes called peptidylarginine deiminases (PADs). Deimination is implicated in a growing number of physiological processes (innate and adaptive immunity, gene regulation, embryonic development, etc.) and concerns several human diseases (rheumatoid arthritis, neurodegenerative diseases, female infertility, cancer, etc.). Here, we update the involvement of PADs in both the homeostasis of skin and skin diseases. We particularly focus on keratinocyte differentiation and the epidermal barrier function, and on hair follicles. Indeed, alteration of PAD activity in the hair shaft is responsible for two hair disorders, the uncombable hair syndrome and a particular form of inflammatory scarring alopecia, mainly affecting women of African ancestry.

## 1. Introduction

Deimination or citrullination corresponds to the conversion of the amino acid arginine within a peptide sequence into the non-standard amino acid citrulline. Deimination is distinct from the formation of free citrulline through the action of nitric oxide synthases on free arginine. Instead, it is a post-translational modification of proteins, catalyzed by a family of calcium-dependent enzymes, called peptidylarginine deiminases or protein-arginine deiminases (PADs, EC3.5.3.15). The number of PAD isotypes increases during evolution from one in fishes and amphibians, three in reptiles and chicken, to five in mammals, including human (called PAD1, 2, 3, 4 and 6, encoded by the corresponding *PADI*1-4 and 6 genes) [1,2,3]. The paralogous *PADI* genes are clustered on a single chromosome, for example on chromosome 1 in a 355 kb-long locus in position p36.1 in human. PADs replace the primary ketimine group of arginine (=NH) by a ketone one (=O) and yield ammonia as a side-product (Figure 1a). Since arginine is positively charged at a neutral pH, whereas, citrulline is not, deimination induces a decreased net charge of targeted proteins; this can change their hydrophobicity, folding, and intra- or inter-molecular ionic interactions, leading to changes in their function and their fate.

Deimination is involved in a growing number of physiological processes (innate and adaptive immunity, control of gene expression, embryonic development, etc.) and has been associated with several human diseases (cancer, rheumatoid arthritis, neurodegenerative diseases, etc.) [4,5,6,7,8].

PADs are 663–665 amino acids long proteins with a molecular mass of ~74 kDa, except PAD6 that contains 694 amino acids [1]. The three-dimensional structure of PAD1–4 has been obtained using X ray crystallography [9,10,11,12] or in silico modeling [13]. They are formed by two immunoglobulin-like N-terminal subdomains from Met1 to Pro300 fused to a highly conserved C-terminal domain that contains the active site cleft (Figure 1b,c). One histidine, two aspartic acids and one cysteine are necessary for the enzymatic activity (Figure 1b,c). PAD2–4 were shown to form head-to-tail homodimers whereas PAD1 seemed to be monomeric (Figure 1d). [9,10,11,12]. In addition, PAD activity and substrate recognition requests four (PAD1) to six (PAD2) calcium-binding sites, located along the amino acid sequence (Figure 1c,d). Calcium binding is required for the enzymatic effectiveness and leads to structural changes that generate the active site [9,14]. During the deimination reaction, an essential cysteine (Cys645 in PAD1 and PAD4, Cys647 in PAD2, Cys646 in PAD3; UniProtKB accession numbers Q9ULC6, Q9Y2J8, Q9ULW8, and Q9UM07, respectively) reacts with the guanidino group of the targeted arginine. A covalent tetrahedral intermediate is formed with release of ammonia. Finally, following adduct hydrolysis, the cysteine is regenerated and the keto-group formed [9,13]. PAD4 was also shown to act on mono-methyl-arginyl residues of histones, yielding to citrullyl residues and methylamine [15].

The expression and activity of PADs are regulated at multiple levels, including transcriptional, translational and post-translational levels (for a review see [13]). For example, 1-α, 25-dihydroxyvitamin D3 induces the expression of PAD mRNA in keratinocytes [16,17], and auto-deimination of PADs changes their tertiary structure and interferes with their enzymatic activity or protein-protein interactions [16,18,19]. The first and so far only biological regulator of PAD, namely the tyrosine-protein phosphatase non-receptor type 22 (PTPN22), was recently characterized as a non-enzymatic inhibitor of PAD4 [20]. Mouse Pad3 was demonstrated to be S-palmitoylated on five cysteines (Figure 1c,d), palmitoylation being critical for the protein stability [21]. Whether PAD3 palmitoylation is involved in its localization or is major to control its activity remains to be tested. Since three out of the five cysteines, shown to be palmitoylated in mouse Pad3, are conserved in the other human isotypes, except PAD6 and orthologous enzymes, we speculate that they could be modified as well.

PADs display unique patterns of tissue expression and substrate specificity, and thus of function. For more than twenty years, we are interested in the role of PADs in the skin. We have identified several substrates in the epidermis and hair follicles, including keratins and S100-fused type proteins, and have characterized the effects of their deimination. We also identified two hair disorders due to mutations in the PAD3 encoding gene: A rare hair shaft dysplasia called uncombable hair syndrome (UHS; OMIM#191480, #617251, and #617252) [22], and a particular form of alopecia, the Central Centrifugal Cicatricial Alopecia (CCCA, OMIM%618352), affecting up to 5% of women of African ancestry and the most common type of primary scarring alopecia [23]. 

## 2. Expression and Role of PADs in Normal Skin

The expression of PADs in the skin has been investigated using RT-PCR analysis and isotype specific antibodies: only PAD1, PAD2, and PAD3 were detected, mainly in the epidermis and some skin appendages [1,24,25,26,27]. In addition, expression of Pad4 has been detected in rat epidermis [28]. In parallel, deiminated proteins have only been immune-detected in the *stratum corneum*, the most upper layer of the epidermis, and in hair follicles [24,25,26,27]. 

### 2.1. PADs and Deiminated Proteins in the Epidermis

PAD1 is present in all layers of the epidermis from the basal layer to the superficial stratum corneum with an increasing expression from the basal to the granular keratinocytes. It is located in the cytoplasm of keratinocytes and in the intracellular filamentous matrix of corneocytes. In the granular keratinocytes it is particularly found in keratohyalin granules and on keratin intermediate filaments (Figure 2a–c). PAD2 is expressed by all keratinocytes, from a very low level in the basal cells to a high level in granular cells where it is located at the cell periphery. PAD3 is specifically expressed by granular keratinocytes where it is located in keratohyalin granules (Figure 2d,e), and persists in the filamentous matrix of only the lower corneocytes. The fact that deiminated proteins are only detected in the stratum corneum (Figure 2f) suggests that PADs are activated in the last step of keratinocyte differentiation, at the transition between granular to cornified layers, where the calcium concentration is the highest. Another possibility is that, lightly deiminated proteins or deiminated proteins, expressed at a low level, remain to be discovered.

After chemical modification of citrullines in strong acidic conditions in the presence of diacetyl monoxime and antipyrine, and using the AMC antibody, an antibody specifically directed against deiminated proteins, four PAD targets have been identified by immuno-blotting: Keratins KRT1, and KRT10, and two members of the S100-fused protein family named filaggrin and hornerin [29,30,31,32]. A third member of the family, filaggrin-2, has also been strongly suspected to be deiminated [33]. All these proteins are covalently cross-linked by transglutaminases (TGases) to the cornified envelope [34], a singular resistant structure that replaces the plasma membrane at the periphery of differentiated keratinocytes. We demonstrated that cornified envelopes indeed contain deiminated proteins, suggesting they are cross-linked after their deimination [32]. We therefore asked whether deimination may favor their binding. We showed that deimination of a recombinant form of hornerin improved intra- and inter-chain cross-linking by both TGases 1 and 3 in in vitro assays [32]. We also analyzed by mass-spectrometry cornified envelopes purified from plantar epidermis obtained from two healthy volunteers [35], using deimination of arginine residues and deamidation of glutamine and asparagine as possible post-translational modifications. Detection of peptides containing only arginine(s) as potentially modified amino acid(s) allowed us to confirm without any ambiguity that the cornified envelopes contain deiminated KRT1, filaggrin, hornerin, and filaggrin-2, and to map some of the citrulline positions (unpublished data; Table 1). In addition, we identified additional deiminated cross-linked keratins, namely KRT2, KRT14, and the hyperproliferation-related keratins KRT6, KRT16, and KRT17. The deiminated arginines in human keratins, as suggested in mouse keratins [36], are located in the non-helicoidal head (V1 subdomain) and tail (V2) domains. The functional importance of this observation is unknown. The data also pointed out the deimination of other proteins that were not previously shown to be deiminated, including the keratinocyte proline-rich protein (Table 1).

Besides their role as cornified envelope components, filaggrin and filaggrin-2 are two key proteins for the epidermal barrier function, and deimination is a major step in their similar complex metabolism [34,37,38,39]. Indeed, filaggrin and filaggrin-2 are two basic proteins that interact with and promote the aggregation of keratin intermediate filaments leading to the formation of the intracellular corneocyte matrix. Then, filaggrin and filaggrin-2 are deiminated by PAD1 and/or PAD3. This reduces their positive charge, decreases their affinity for keratins and induces their dissociation from the matrix. Subsequently, a small amount of filaggrin and the amino-terminal part of filaggrin-2 (including the S100-like, the spacer and the A-type repeat domains) are covalently bound to cornified envelopes, deimination increasing the efficacy of cross-linking. The major part of filaggrin and the carboxy-terminal part of filaggrin-2 (the B-type repeat domain) are fully degraded to generate free amino acids. Deimination improves the efficacy of proteolysis, at least in vitro, by bleomycin hydrolase and calpain 1 [33,40]. These amino acids and some of their derivatives, including urocanic acid, that is derived from histidine, and pyroglumate, which is derived from glutamine, participate to the formation of the Natural Moisturizing Factor, a pool of hygroscopic molecules that collectively retain water in the upper stratum corneum. In addition, trans-urocanic acid absorbs part of ultraviolet B-radiations, acting as a natural sun screen [41]. PAD1 expression and activity appears to be a master regulator of filaggrin degradation. Indeed, when reconstructed 3D human epidermis is produced at low relative humidity (~30%; dry conditions), expression of PAD1 is largely induced, and filaggrin deimination is up-regulated in the lower stratum corneum (Figure 3a,b). This leads to higher amount of urocanic acid and pyroglutamate [42]. Fifty percent inhibition of PADs using chloro-amidine, a pan-PAD inhibitor non-toxic for keratinocytes, partly reversed this consequence of dryness on keratinocyte metabolism [42]. Similarly, inhibition of PADs in an organotypic model, derived from rat palate keratinocytes reduced the breakdown of FLG and increased its association with keratins [43]. In this experimental model, activation of PADs with acefylline, a PAD1 and PAD3 activator [44], had the opposite effects. Accordingly, deimination and degradation of filaggrin occur immediately after birth of newborn rodents, whereas, the application of occlusive patches onto the skin or keeping the animals in a water-saturated environment, prevents this effect due to passage from amniotic liquid to air [45,46].

To understand whether PADs play a wider role in keratinocyte biology and epidermal homeostasis, the phenotype of reconstructed human epidermis produced with normal keratinocytes and treated with chloro-amidine was analyzed in more details [47]. Cornification, the transformation of keratinocytes to corneocytes, was shown to be attenuated, and the associated autophagy process to be altered with an increased number of transitional cells and a corresponding reduced number of corneocyte layers, and with ultrastructural alterations in the cytoplasm of granular keratinocytes, including clustering of mitochondria, the presence of autophagosomes, and the accumulation of heterogeneous vesicles, sometimes close to the nucleus [47] (Figure 3c,h). In agreement, markers of autophagy such as LC3B, sestrin-2 and sequestosome-1 were detected in higher amount. In addition, LC3B was shown to partly colocalised with PAD1 and PAD3. These data strongly suggest that deimination of proteins regulates autophagy in keratinocytes. In agreement with this hypothesis, PAD4 is known to be associated with autophagy in various cell types. For examples, the autophagy flux in a human osteosarcoma cell line has been found to be regulated by the PAD4-specific inhibitor YW3–56 [48], and the rapamycin-mediated induction of autophagy in fibroblasts leads to PAD4 activation [49]. Deciphering the respective role of PAD1 and PAD3 in the cornification-related autophagy process remains to be done.

The specific role of PAD2, if any, in the epidermis is not known. When the skin of mice which Padi2 gene has been inactivated were analyzed, no phenotype was observed. There was neither any effect on skin wounding [50]. The fact that PAD2 is located at the periphery of granular keratinocytes [25], where cornified envelopes are formed, suggests an involvement of this isotype in the deimination of cornified envelope components, but this has not yet been investigated.

### 2.2. PADs and Deiminated Proteins in Hair Follicles

PAD1, PAD2, and PAD3 were also immune-detected in most skin appendages, except sebaceous glands [26,27]. PAD1 is present in secretory and myoepithelial cells of sweat glands, in arrector pili muscular cells and in hair follicle keratinocytes. In the hair follicles, the anti-PAD1 antibody stained the Huxley layer, part of the inner root sheath, as well as the companion layer. PAD2 is only expressed in secretory and myoepithelial cells of sweat glands, and in the muscle cells of arrector pili, but it was not detected in the hair follicles. However, PAD3 is detected in the hair follicles, in particular, the three epithelial layers of the inner root sheath (i.e., the Henle layer, the Huxley layer, and the cuticle), and both the cuticle and medulla of the hair shaft, but not in any other appendages (Figure 4a). Using an antibody directed against deiminated proteins, no signals were detected in skin appendages except the three layers of the inner root sheath and both the medulla and cuticle of hair follicles (Figure 4b) [26,27].

Until now, few deiminated proteins have been identified in the hair follicles. The first one was trichohyalin, another member of the S100-fused type protein family, 1943 amino-acids long, which is abundant in the inner root sheath and medulla [34,41]. Trichohyalin plays an essential role in hair morphology. Indeed, a genome-wide association study for genetic polymorphisms, associated with the shape of the hair (stiff, wavy or curly), was carried out using three cohorts of Australians of European origin, showed a strong link (*p* = 3.2 × 10^−31^) between the probability of having straight hair and a polymorphism of the *TCHH* gene encoding trichohyalin (rs11803731; c.2368T>A/*p*.Leu790Met) [51]. These results were confirmed by a second study with two European cohorts [52]. A mass spectrometry analysis of mouse coat also showed the importance of trichohyalin for follicle architecture [53]. Deimination of trichohyalin induces changes in the protein structural organization, and allows the solubilization of cytoplasmic granules where it is aggregated. The future of the protein is then site-dependent: In the inner root sheath it interacts with keratin filaments and is cross-linked by TGase 3 to itself, to keratins, and to other cornified envelope proteins of fully differentiated keratinocytes, providing mechanical strength to the hair follicle sheath and supporting the hair shaft growth; in the medulla, it forms TGase 3 mediated cross-linked homo-oligomers and vacuolated aggregates that appears to entrap air, being important for thermal insulation [31,41,54]. The only PAD isotype, expressed in the medulla and the Henle’s layer, is PAD3. Therefore, it is this isotype that is responsible for the deimination of trichohyalin, at least in this part of the hair. In agreement, PAD3 and trichohyalin were shown to co-localize there (Figure 4c) [26]. 

Several keratins of the hair follicles are also substrate of PADs, including KRT25, KRT2, KRT28 and KRT71 [31,54]. Finally, S100A3, a small (11 kDa) calcium- and zinc-binding protein highly expressed in the hair cuticle and supposed to reinforce the cuticle is deiminated by PAD3. S100A3, a protein predicted to form a dimer, contains only four arginine residues, and it is mainly modified on arginine 51. In vitro PAD3-catalyzed deimination of S100A3 results in its assembly as a homotetramer, and improved its calcium binding ability [27,55,56]. Intriguingly, the genes encoding S100A3 and PAD3 seem to have co-evolved during evolution of mammals, since, for example, there is co-loss-of-function in cetaceans mostly lacking hair follicles [3].

## 3. PAD Defects and Skin Diseases

Although, more data show the importance of PADs in skin physiology, few data are available concerning their direct implication in skin diseases, except for two hair disorders [22,23]. 

Lower deimination of keratins, in particular KRT1, was reported in the lesional skin of psoriatic and atopic dermatitis patients, as well as in the epidermis of patients with bullous congenital ichthyosiform erythroderma [57,58,59]. The expression of PADs appeared to be normal in the epidermis of the former patients, suggesting altered activity. In skin carcinomas and extra-mammary Paget’s disease, ectopic expression of PAD4 was reported [60,61]. This suggest the involvement of PAD4 in skin tumorigenesis, since the inhibition of PAD4 results in cell cycle arrest and apoptosis [62], and given PAD4 is involved in the repression of p53 target genes [63]. In a genome-wide study of two cohorts of patients with basal cell carcinoma, the most common cancer among Europeans, a single nucleotide polymorphism in *PADI*6 gene was highlighted, with an estimated risk of 2.68 [64]. It is most likely that alteration in the expression and activity of PADs, in the epidermis of patients with inflammatory and other skin diseases, will be explored in a near future.

In 2016, in an international collaborative work, we discovered that nine children affected by a condition known as UHS, coming from nine different families, carry homozygous or compound heterozygous mutations in the *PADI*3 gene (c.881C>T/p.Ala294Val; c.335T>A/p.Leu112His; c.1813C>A/p.Pro605Thr) (Figure 5a–e, i) [22]. UHS, also known as ‘pili trianguli and canaliculi’, is a rare hair shaft dysplasia that appears in young children and generally improves with age. This hair disorder is characterized by dry, curly, and shiny hair, with so tousled follicles that grow in multiple directions and are impossible to flatten on the scalp (Figure 5a). In some cases this hair anomaly has been described in association with other diseases, such as ectodermal dysplasia. The analysis of the hairs by scanning electron microscopy, the best diagnostic test, reveals a longitudinal groove along the length of the hair shaft with a triangular- or kidney-shaped section (Figure 5b). UHS is mostly inherited in an autosomal recessive manner, although cases inherited in an autosomal dominant fashion, with partial penetrance, have been reported. 

Functional studies, including the expression of recombinant forms of wildtype and mutated PAD3 (produced after site directed mutagenesis) in bacteria and cultured human keratinocytes, as well as in silico modeling of the enzyme structure, showed that UHS-responsible mutations induce structural changes and a marked decrease in the activity of the corresponding enzyme. In addition, characterization of 7-weeks-old *Padi*3 knockout mice revealed alteration in the morphology of mouse hair coat and whiskers, with a rough and irregular surface that appears hammered (Figure 5c,d) [22]. In an additional Turkish family, we showed UHS to be caused by a homozygous non-sense mutation (c.1351C>T/p.Gln451X) in the *TGM*3 gene encoding TGase 3. Similar functional experiments showed that the mutation led to a shorter protein with a reduced enzymatic activity. Finally, a loss-of-function mutation of the trichohyalin encoding gene (c.991C>T/p.Gln331X) was identified in an 11th family [22]. This mutation results in the synthesis, if it takes place, of a very short form of trichohyalin, probably unable to both interact with keratins and form cross-linked complexes, necessary for its normal function in the inner root sheath, and hair shaft, respectively. Accordingly, mice with an inactivated *Tgm*3 gene have twisted vibrissae and a wavy coat, with fragile follicles related to major changes in cuticle cells [65]. 

In 2019, we reported in the New Englang Journal of Medicine [23] results of a genetic study of a condition known as CCCA, which implicates variants in *PADI*3. CCCA is a common and progressive type of scarring inflammatory alopecia affecting predominantly female of African ancestry, with an estimated prevalence of 2.7 in women of South Africa and 5.6% in Afro-American women in the USA [66,67,68]. Hair-breakage and thinning at the vertex scalp progress centrifugally. Histopathological examinations of skin biopsies reveal perifollicular lymphocyte infiltration and fibrosis [69], and follicular degeneration (Figure 5f,g). The disease is associated with traction-inducing hair-grooming practices and usage of hair-irritant chemicals. As several members of the same family can be affected, some without any history of hair trauma, the primary pathologic event has been implicated as a genetic defect, with environmental stressors considered as exacerbation factors [66,67,68]. Exome sequencing of 16 women with CCCA yielded one splice-site mutation (c.832-2A>G; skipping of exon 8, which in turn leads to a frameshift-predicted mutation expected to replace amino acids 278–664 by 38 unrelated amino acids due to the frameshift) and three heterozygous missense mutations in *PADI*3 (c.856A>G/p.Thr286Ala; c.1669C>T/p.Arg557Trp; c.1744G>A/p.Ala582Thr; Figure 5h,i) for five patients [23]. Protein modeling suggested that these mutations result in protein mis-folding. Functional experiments, similar to those described for UHS mutation analysis, revealed reduced stability and induced aggregation of the mutated proteins. RNA sequencing data and RT-qPCR experiments showed a reduction in the mRNA level of many proteins, known to be involved in hair formation, including trichohyalin and S100A3, in the scalps of affected patients. Replication of the genetic analysis in a cohort of additional 42 patients with CCCA identified *PADI*3 mutations, including two new mutations (c.1955G>A/p.Arg652Lys; c.628C>T/p.Arg210Trp), in 9 of them (Figure 5h,i). Altogether, six different heterozygous mutations of *PADI*3 were identified in 14 out of 58 patients (24%) with CCCA [23]. Interestingly but curiously, these mutations are distinct from those mutations responsible for UHS (Figure 5i), suggesting different pathogenic consequences. However, *PADI*3 mutations in children with UHS occur in either a homozygous or a compound heterozygous pattern, whereas in patients with CCCA mutations are heterozygous. In addition, the CCCA mutations but not the UHS mutations may change the location of the encoded enzyme or its substrate selectivity. Finally, it is not unique that different mutations of one particular gene induce two different clinical skin phenotypes. For example, different non-sense mutations of the *CDSN* gene, encoding corneodesmosin, are responsible for either, peeling skin syndrome type 1 (OMIM#270300) [70] or hypotrichosis simplex of the scalp (OMIM#146520) [71]. 

As a result of these findings, growing evidence is demonstrated for the essential role of the trichohyalin-PAD3-TGase 3 pathway in the physiology of the hair follicle and hair shaft formation.

## 4. Conclusions

In conclusion, PADs are increasingly important enzymes in many aspects of skin physiology and in human diseases, particularly in the epidermal barrier function, hair shaft formation and hair diseases, as summarized below in five points:In skin, PADs are regulated at multiple levels, including gene expression, mRNA translation, and post-translational modifications, such as autodeimination and S-palmitoylation;Cornified envelopes contain deiminated proteins, including filaggrin, hornerin, and the newly identified keratinocyte proline rich protein, deimination improving the efficacy of cross-linking by transglutaminases;PAD inhibition using chloro-amidine slows down cornification and alters the associated autophagy process in the granular keratinocytes, showing the importance of these enzymes in the last steps of keratinocyte differentiation;As demonstrated by the comparison of reconstructed human epidermis produced in dry versus humid atmospheric conditions, keratinocyte external environment modifies PAD1 expression and deimination, and the subsequent filaggrin proteolysis;The direct implication of PAD3 gene mutations has been demonstrated in two human hair disorders, a rare hair shaft dysplasia and a common scarring inflammatory alopecia.

## Figures and Tables

**Figure 1 ijms-21-00566-f001:**
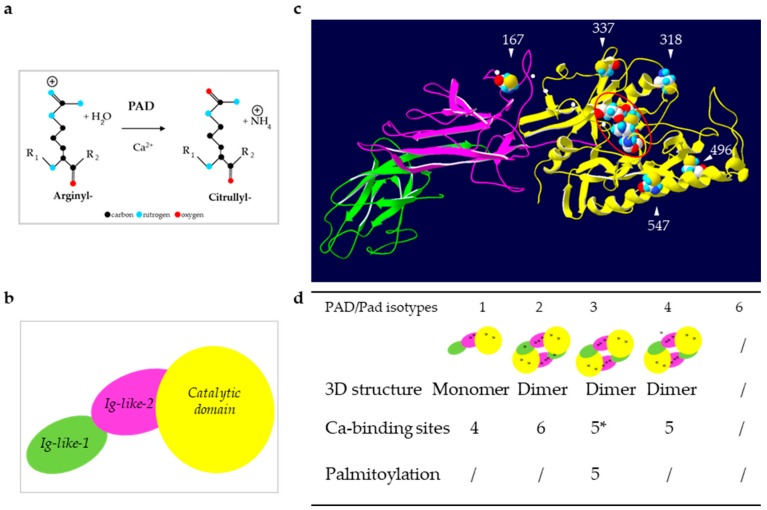
Reaction of deimination and structure of PADs. (**a**) Schematic representation of the reaction catalyzed by PADs: Deimination or citrullination. (**b**) Schematic representation of the sub-domains of PADs. (**c**) Illustration of an in silico three-dimensional (3D) model of the active PAD3. The white arrowheads indicate the five putatively palmitoylated cysteines (by similarities to mouse Pad3). The small white dots indicate the five conserved calcium binding sites. The four gathered major amino-acids of the active site are highlighted by a red oval (Asp350, His470, Asp472 and Cys646 by similarities to PAD4). (**d**) Summary of the structural data for each human (PAD) and mouse (mPad) isotypes. Positions of the calcium binding sites are indicated by black dots on each sub-domain representation. * As observed after a multiple sequence alignment (MultAlin), the amino-acids involved in the five calcium binding sites are highly conserved, especially between PAD4 and PAD3 [13]. / means that the information is not known.

**Figure 2 ijms-21-00566-f002:**
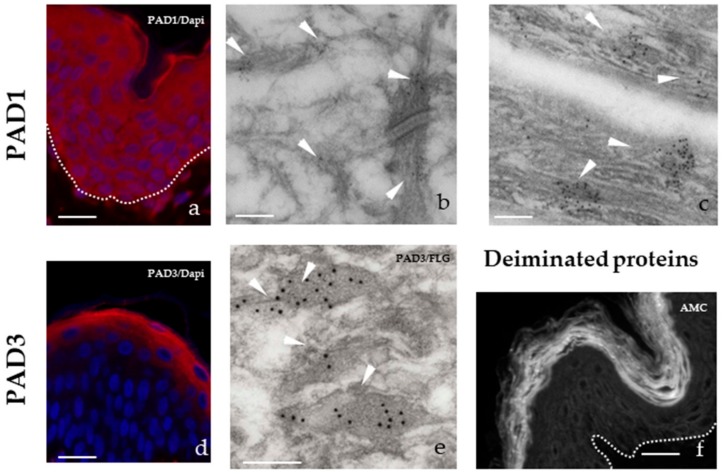
PADs and deiminated proteins in normal human abdominal epidermis. (**a**) Immunofluorescence detection of PAD1 (red). (**b** and **c**) Immuno-electron microscopy detection of PAD1. White arrowheads point gold beads on intermediate filaments in a granular keratinocyte (**b**) and on the intracorneocyte matrix (**c**). (**d**) Immunofluorescence detection of PAD3 (red). (**e**) Double labelling of PAD3 (small gold beads) and filaggrin (large beads) on keratohyalin granules (arrowheads) in a granular keratinocyte. (**f**) Immunofluorescence detection of deiminated proteins using the AMC antibody specific for chemically modified citrullines, as previously described [29]. The dermo-epidermal junction is indicated by a doted white line (**a** and **d**). Bar = 50 µm (**a**, **d** and **f**) and 200 nm (b, c, and e). Nuclei are stained in blue (Dapi) (**a** and **d**).

**Figure 3 ijms-21-00566-f003:**
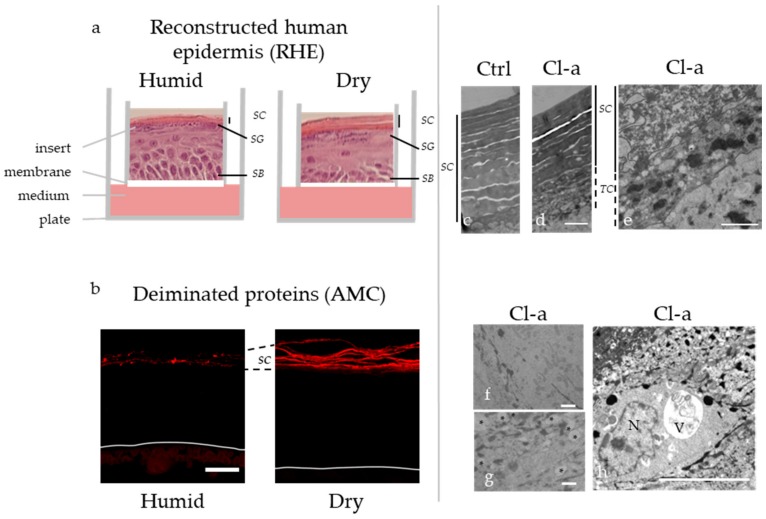
Production of 3D reconstructed human epidermis (RHE) in a dry atmosphere induces deimination while inhibition of deimination by Cl-amidine alters autophagy at the transition *stratum granulosum / stratum corneum.* (**a**) Schematic representation of the RHE produced in a humid (relative humidity = 95–100%) *versus* dry (relative humidity around 30%) atmosphere, with hematoxylin-eosin staining of a representative RHE section in each condition. (**b**) Immunofluorescence detection of deiminated proteins with the AMC antibody in RHE produced in a humid or dry atmosphere. Bar = 20 µm. (**c**–**h**) Transmission electron microscopy observations of control RHE (Ctrl), and RHE treated with Cl-amidine (Cl-a). The corneocytes and transitional cells are delineated by vertical full or dotted line, respectively (**c**–**e**). Clustered mitochondria are shown (**f**) and heterogeneous small vesicles are pointed by asterisks (**g**). A large vesicle (V) close to a nucleus (N) with a non-ovoid irregular shape is also illustrated (**h**). Bar = 1 µm. SB: *stratum basale*, SC: *stratum corneum*, SG: *stratum granulosum*, TC: transitional cells.

**Figure 4 ijms-21-00566-f004:**
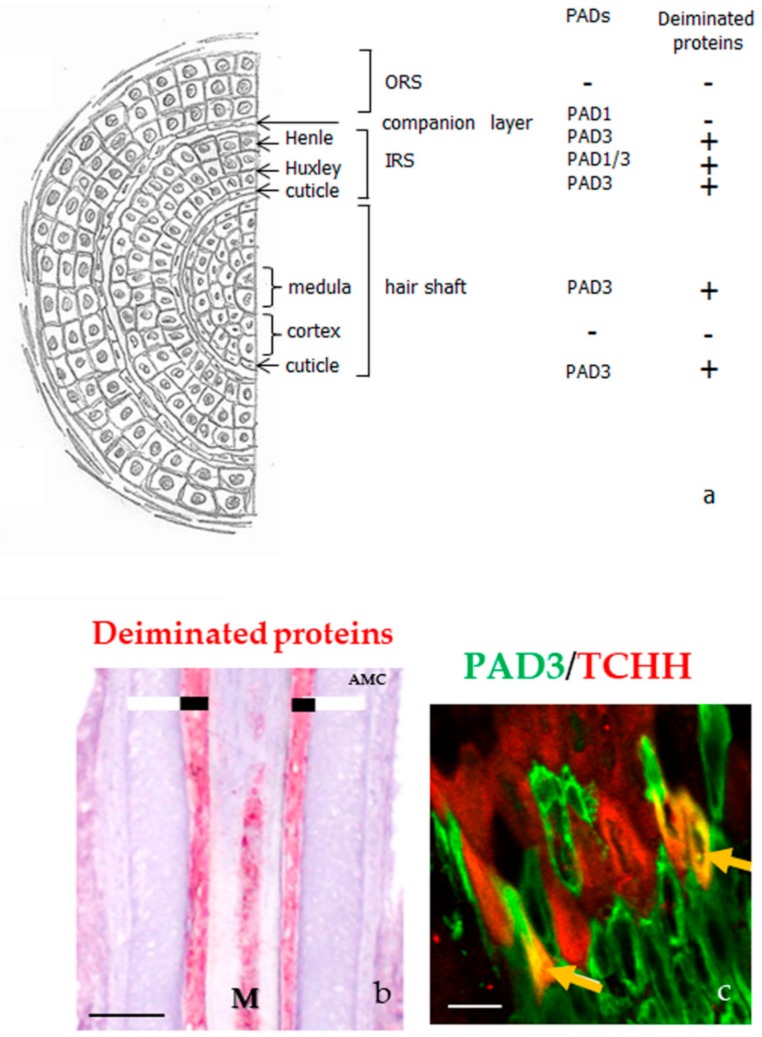
PADs and deiminated proteins in the hair follicles. (**a**) Schematic representation of a hair follicle section and detection of both PADs and deiminated proteins, as reported in [26,27]. IRS, inner root sheath; ORS, outer root sheath. (**b**) Immunodetection of deiminated proteins (red) with the anti-modified citrulline antibody in the medulla (M) and the inner root sheath (large black line) of a normal human hair follicle. The outer root sheath is indicated with a large white line. (**c**) Immunofluorescence detection of PAD3 (green) and trichohyalin (TCHH, red). A yellow staining (yellow arrows) corresponds to colocation of the two proteins. Bar = 50 µm (**b**) and 100 µm (**c**).

**Figure 5 ijms-21-00566-f005:**
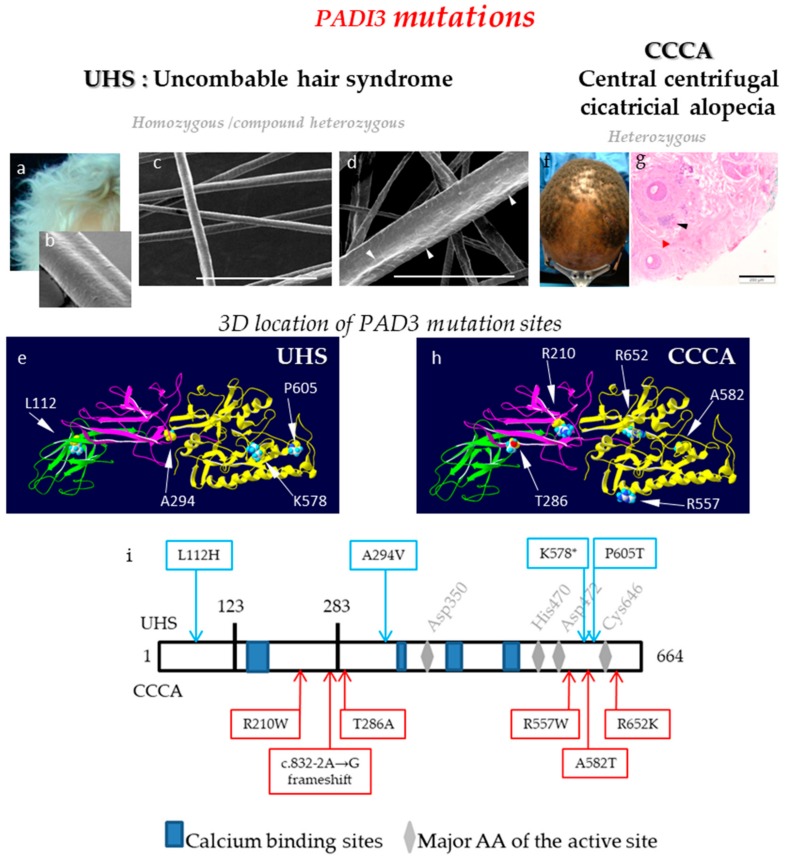
PADs and deimination in hair follicle disorders. (**a**) Illustration of the macroscopic hair phenotype of an UHS patient. (b-d) Representative scanning electron microscopy observations of the abnormal longitudinal UHS hair shape (**b**), hair follicles of *Padi3* wild type mice; (**c**) and hair follicles of *Padi*3 knockout mice (**d**). Bar = 150 µm. White arrowheads point the abnormalities of the hair shape. (**e**) Localization of PAD3 amino-acids that are mutated in UHS patients. (**f** and **g**) Hair phenotype of a Central Centrifugal Cicatricial Alopecia (CCCA) patient and histological observation. Black and red arrowheads point to inflammatory infiltrate, and fibrosis, respectively. Bar = 200 µm. (**h**) Localization of PAD3 amino-acids that are mutated in CCCA patients. (**i**) Position of the pathological mutations on a linear representation of PAD3. Mutations responsible for UHS (top) or CCCA (bottom). The in silico 3D model of PAD3 (**e**,**h**) was produced by similarity from the crystallographic data of PAD4 (PDB ID: 1WDA), as previously published [9,16,22].

**Table 1 ijms-21-00566-t001:** Deiminated proteins identified by mass-spectrometry analysis of purified human plantar cornified envelopes.

Protein Identifier ^1^	Protein Name (Abbreviation)	Deiminated Arg Number
P04264	Keratin 1 (KRT1)	R65, R82
P35908	Keratin 2 (KRT2)	R603
P04259	Keratin 6B (KRT6B)	R59, R86
O43290	Keratin 14 (KRT14)	R510
P08779	Keratin 16 (KRT16)	R41
Q04695	Keratin 17 (KRT17)	R41, R191
P20930	Filaggrin (FLG)	R1026 or R3295, R1567
Q5D862	Filaggrin-2 (FLG2)	R637 or R789
Q86YZ3	Hornerin (HRNR)	R1953
Q5T749	Keratinocyte proline-rich protein (KPRP)	R409
P53611	Geranylgeranyltransferase (GGT)	R184
Q7Z4L5	Tetratricopeptide repeat protein 21B (TTC21B)	R297, R301

^1^ UniProtKB/SwissProt accession number.

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
