# Peer review of "Deimination and Peptidylarginine Deiminases in Skin Physiology and Diseases"

_ijms, 2020, doi:10.3390/ijms21020566_

Round 1

Reviewer 1 Report

I have reviewed the manuscript by Mechin et al and recommend to accept it for publication in IJMS with minor revisions, as listed below:

1) pg 4, line 84: change to: “For example,”

2) pg 4, line 93: change to: “PAD3”

3) pg 11, line 249: replace “the” with “an”

4) pg 12, Fig 4: add a space between the words “companion layer” in the figure.

5) pg 13, lines 284-286: Rewrite sentence. Suggestion: “In vitro, PAD3-catalyzed deimination of S100A3 was predicted to form a dimer, but instead resulted in its assembly as a heterotetramer, improving its calcium binding ability [27,55,56].”

6) pg 13, lines 287-288: Rewrite sentence. Suggestion: “… during evolution of mammals, since, for example, there is co-loss-of-function …”

7) pg 14, line 294: change to: “atopic dermatitis patients,”

8) pg 14, line 305: change to: a condition known as Uncomfortable Hair Syndrome (UHS), coming from …”

9) pg 14, line 312: omit “and this is actually”

10) pg 14, line 312: change to: “diagnostic test,”

11) pg 14, line 315: change to: “… dominant fashion, with partial penetrance, …”

12) pg 16, line 329: add a comma after “keratinocytes”

13) pg 16, line 333: omit the word “as”

14) pg 16, line 339: change to: “… unable to both interact with keratins and form …”

15) pg 16, line 341: change to: “… with an inactivated … and a wavy …”

16) pg 16, lines 343-344: change to “… results of a genetic study of a condition known as Central Centrifugal Cicatricial Alopecia (CCCA), which …”

17) pg 16, line 350: change to: “… members of the same …”

18) pg 17, line 352: change to: “… been implicated as a genetic …”

19) pg 17, line 354: change to: “… ; skipping of exon 8, which in turn leads to a frameshift-predicted mutation …”

20) pg 17, line 358: change to: “… mutations result in …”

21) pg 17, lines 361-362: change to: “… S100A3, in the scalps of affected patients.”

22) pg 17, line 368: Rewrite concluding sentence. Suggestion: “ As a result of these findings highlighted here, growing evidence for the essential … formation is demonstrated.”

Author Response

We thank the Reviewer for his(her) comments.

We changed the manuscript according to all the minor points raised, except point 2). Indeed, it is a general rule to note human genes/proteins in capital letters and mouse genes/proteins in lower case letters. However, if this is not a guideline of The Journal, we agree to change “Pad3” to “PAD3”, up to the Editor.

Reviewer 2 Report

In this manuscript, recent progress in the studies on peptidylarginine deiminases (PADs) including the author's own works and other groups' has been summarized with special focus on 1) the expression patterns of PADs, and 2) PAD defects-related skin diseases. The manuscript was well constructed, informative, and understandable for readers even with little background. But a little more descriptions may be needed for better understanding. The details are as follows. 

1) In Fig. 2.  some explanations about "AMC antibody" is required in the figure legend. In addition, it is necessary to indicate some references in which the specificity of the antibody was confirmed.

2) Table 2 would be better if it includes reference numbers.

3) In Fig. 5 legend, add a PDB accession number for the crystal structure of PAD3. 

4) It is curious that two different diseases, UHS and CCCA, are associated with mutations on a single protein. It would be necessary to add a little more comments on possible reasons for the different consequences. 

5) It seems necessary to add a short conclusion section that cover the whole contents of the manuscript. 

Author Response

We thank the Reviewer for his(her) positive comments.

We changed the manuscript as requested, see the point-by-point answer below:

We added some explanations about AMC antibody in the legend of Figure 2 as well as a reference where its specificity has been demonstrated. We also add additional words in the text page 6, line 146. The Reviewer probably talked about Figure 4a. We added two references in the legend of this Figure. The in silico 3D model of PAD3 was produced by similarity from the crystallographic data of PAD4, as previously published. We added this information in the legend of Figure 5, as well as the PDB accession number for the corresponding crystal structure (PDB ID: 1WDA) and the 3 referred publications. We commented this point in a new paragraph, page 17, lines 381-387; and added two references [70,71]. We added a new paragraph “4. Conclusion”, including the 5 major points reported in our manuscript (page 18, lines 399-415).